  

# Redundant and nonredundant organismal functions of EPS15 and EPS15L1

Cinzia Milesi[1], Paola Alberici[1], Benedetta Pozzi[1], Amanda Oldani[1,2], Galina V Beznoussenko[1] , Andrea Raimondi[3], Blanche Ekalle Soppo[1,4], Stefania Amodio[1,4], Giusi Caldieri[1,4,5], Maria Grazia Malabarba[1,4,5], Giovanni Bertalot[4], Stefano Confalonieri[1,4], Dario Parazzoli[1,2], Alexander A Mironov[1], Carlo Tacchetti[3,6] , Pier Paolo Di Fiore[1,4,5], Sara Sigismund[1,4,5] , Nina Offenhäuser[1,2]

**EPS15 and its homologous EPS15L1 are endocytic accessory proteins. Studies in mammalian cell lines suggested that EPS15 and EPS15L1 regulate endocytosis in a redundant manner. However, at the organismal level, it is not known to which extent the functions of the two proteins overlap. Here, by exploiting various constitutive and conditional null mice, we report redundant and nonredundant functions of the two proteins. EPS15L1 displays a unique nonredundant role in the nervous system, whereas both proteins are fundamental during embryo development as shown by the embryonic lethality of -Eps15/Eps15L1-double KO mice. At the cellular level, the major process redundantly regulated by EPS15 and EPS15L1 is the endocytosis of the transferrin receptor, a pathway that sustains the development of red blood cells and controls iron homeostasis. Consequently, hematopoietic-specific conditional Eps15/Eps15L1-double KO mice display traits of microcytic hypochromic anemia, due to a cell-autonomous defect in iron internalization.**

## Introduction

Endocytosis is a process through which cells internalize metabolites, plasma membrane–resident proteins, and signaling receptors, thereby influencing cellular homeostasis. The process is achieved by a variety of entry portals and routes, among which clathrin-mediated endocytosis (CME) is the best characterized one (McMahon & Boucrot, 2011; Sigismund et al, 2012; Kirchhausen et al, 2014; Robinson, 2015). In addition, endocytosis can occur through clathrin-independent routes (non-clathrin endocytosis), whose molecular determinants and cargo specificity are being elucidated (Lundmark et al, 2008; Sigismund et al, 2012, 2013; Boucrot et al, 2015; Renard et al, 2015; Elkin et al, 2016; Caldieri et al,

2017). Besides clathrin, the main actors of CME are the endocytic adaptors, whose main function is to bridge the membrane cargo with the coat. Endocytic adaptor proteins include the clathrin adaptor protein 2 complex (AP2) and a plethora of accessory adaptor proteins, among which are EPS15 (EGFR pathway substrate 15) and EPS15L1 (EGFR pathway substrate 15 like-1), which control cargo selection and maturation of the vesicles (Mettlen et al, 2009; Traub, 2009; Kirchhausen et al, 2014; Merrifield & Kaksonen, 2014).

EPS15 and EPS15L1 share 41% identity and 61% similarity. They were originally discovered as substrates of the epidermal growth factor receptor (EGFR) (Fazioli et al, 1993; Schumacher et al, 1995; Coda et al, 1998; Salcini et al, 1999) and display features of multi-domain scaffolding proteins, containing from N terminus to C terminus: (i) three copies of the EH protein–binding module (Wong et al, 1995; Di Fiore et al, 1997; Confalonieri & Di Fiore, 2002), which can interact with various endocytic proteins (Salcini et al, 1997; Polo et al, 2003); (ii) AP2-binding sites (Benmerah et al, 1996; Iannolo et al, 1997); and (iii) ubiquitin-binding domains, UIMs (Polo et al, 2002). In line with their homology, in vitro studies indicated that EPS15 and EPS15L1 are redundant components of the endocytic machinery: they share the same binding partners (van Bergen En Henegouwen, 2009) and endocytosis is redundantly affected when both proteins are functionally impaired. In particular, RNA interference of EPS15 and EPS15L1 in HeLa cells showed that these proteins are redundantly required for the CME of the transferrin receptor (TfR), a prototype of constitutive endocytosis, and of the EGFR, a prototype of ligand-induced endocytosis (Huang et al, 2004). Moreover, EPS15 and EPS15L1 were found to redundantly regulate also the non-clathrin endocytosis of EGFR, together with the endocytic adaptor epsin-1 (Sigismund et al, 2005).

Finally, upon activation of the EGFR, both proteins can be post-translationally modified, including tyrosine phosphorylation (Fazioli et al, 1993; Coda et al, 1998) and mono-ubiquitination (van Delft et al, 1997; Woelk et al, 2006), and these modifications are required

[1]IFOM, Fondazione Istituto FIRC (Fondazione Italiana per la Ricerca sul Cancro) di Oncologia Molecolare, Milan, Italy   [2]Cogentech Società Benefit Srl, Milan, Italy  [3]Experimental Imaging Centre, Istituto di Ricovero e Cura a Carattere Scientifico, San Raffaele Scientific Institute, Milan, Italy   [4]IEO, Istituto Europeo di Oncologia IRCCS (Istituti di Ricovero e Cura a Carattere Scientifico), Milan, Italy   [5]Università degli Studi di Milano, Dipartimento di Oncologia ed Emato-oncologia, Milan, Italy  [6]Dipartimento di Medicina Sperimentale, Università degli Studi di Genova, Genoa, Italy

Correspondence: nina.offenhauser@cogentech.it; sara.sigismund@ieo.it
Sara Sigismund and Nina Offenhäuser jointly supervised this work.

to assist during EGFR-CME (Torrisi et al, 1999; Confalonieri et al, 2000; Savio et al, 2016).

At the organismal level, studies in *Caenorhabditis elegans* and *Drosophila melanogaster*, where only one gene of the *eps15* family exists, indicate that EPS15 plays an essential function in the nervous system, regulating synaptic vesicle recycling (SVR) (Salcini et al, 2001; Koh et al, 2007). The single *Eps15* KO in mice, where both *Eps15* and *Eps15L1* exist, is viable and fertile. MEFs derived from these mice did not display defects in TfR endocytosis (Pozzi et al, 2012), suggesting that EPS15L1—which is expressed in this setting—possibly compensates the lack of EPS15. Their redundancy, however, has never been demonstrated in in vivo settings. Moreover, neither in vitro nor in vivo studies have so far revealed nonredundant essential functions. The present studies were undertaken to shed light on these questions.

## Results and Discussion

### The *Eps15L1*-KO is perinatal lethal

To investigate the functions of EPS15L1 at the organismal level, we generated *Eps15L1*-KO (*Eps15L1*-KO) mice by deleting the first coding exon (Fig S1). Although deletion of *Eps15*, as previously reported, did not affect viability (Pozzi et al, 2012), only 10% of the weaned pups were *Eps15L1*-KO (instead of the expected 25%, Fig 1A), indicating that EPS15L1 is required for neonatal viability, albeit with incomplete penetrance. *Eps15L1*-KO mice were born at the expected Mendelian ratio and were indistinguishable from WT littermates. Following the litters directly after birth revealed that newborn *Eps15L1*-KO died within the first 2 d of birth (Fig 1B) without any obvious morphological defects. The neonatal lethality of *Eps15L1*-KO mice, at variance with *Eps15*-KO, indicated a unique nonredundant function for EPS15L1.

### EPS15L1 is preferentially expressed in the nervous system and the *Eps15L1*-KO displays neurological deficits

The time of neonatal lethality of *Eps15L1*-KO mice and their normal in utero development was not immediately consistent with a cause of death because of failure of major organs, such as the heart, lungs, or kidneys (Turgeon & Meloche, 2009). Thus, as an initial step towards the understanding of the nonredundant function of EPS15L1, we analyzed its tissue distribution versus EPS15. EPS15 was expressed at similar levels in most of the analyzed tissues (Fig S2A). EPS15L1, instead, displayed prominent expression in the brain and

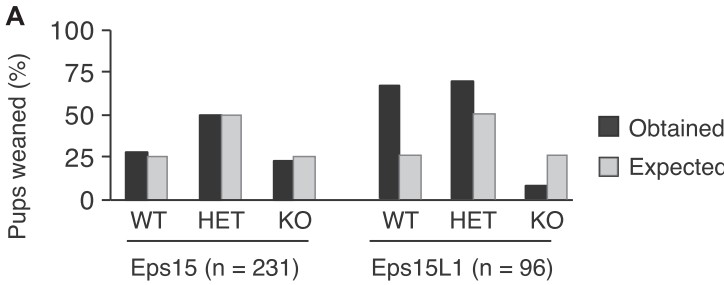
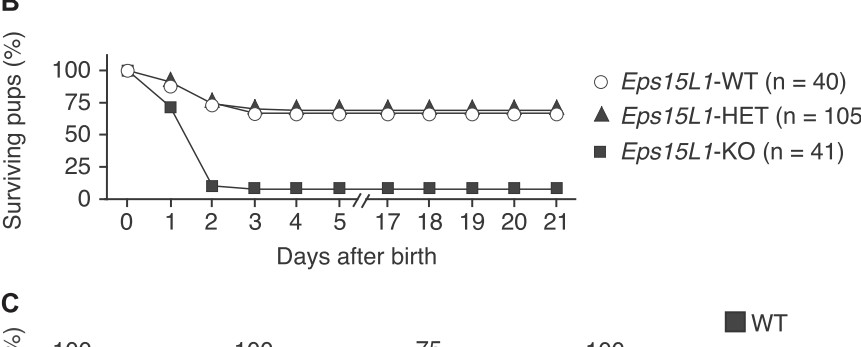
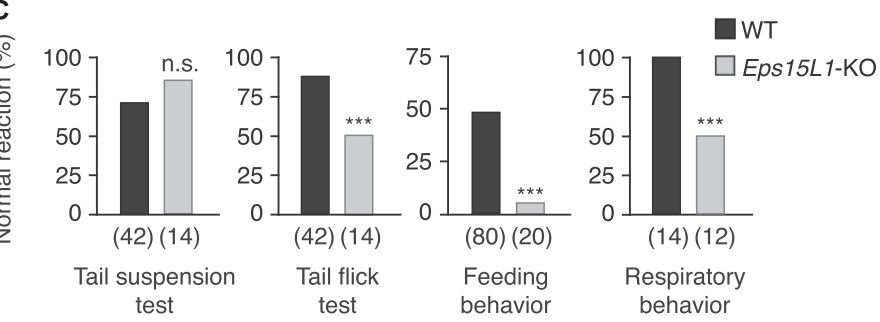

**Figure 1. *Eps15L1*-KO mice are neonatal lethal.**
**(A)** Percentage of expected (black bars) and obtained (grey bars) weaned pups of WT, heterozygous (HET), and KO genotype born from *Eps15*- or *Eps15L1*-HET breedings. **(B)** Percentage of surviving pups of WT, HET, and KO genotype after birth. The number of mice analyzed is shown in parentheses. **(C)** Percentage of normal reaction in the tail suspension, tail flick, feeding behavior, and respiratory behavior tests in *Eps15L1*-WT and *Eps15L1*-KO pups. The number of mice analyzed is shown in parentheses. n.s., not significant; ***$P < 0.001$ versus WT.

cerebellum and comparatively lower levels in other tissues and MEFs (Fig S2B).

By immunofluorescence analysis of the hippocampus in adult mouse brain sections, both EPS15 and EPS15L1 localized to neurons, and EPS15 also showed substantial staining in cells with astrocytic morphology (Fig S2C). Of interest, EPS15L1 showed co-localization with synaptophysin, a presynaptic marker, whereas EPS15 did not (Fig S2C). This was further corroborated by a biochemical fractionation of adult mouse brain in which we compared the total homogenate (H) to the synaptosomal fraction and to the postsynaptic density. Of the several endocytic proteins tested, EPS15L1 was the sole one clearly enriched in the synaptosomal fraction (marginal enrichment was also detected for dynamin and AP2; Fig S2D).

Based on the above results, we performed a series of neurological tests on newborn *Eps15L1*-KO (Fig 1C). We observed no difference in the tail suspension test, which assesses a reflex circuit involving the brainstem (Fig 1C). Conversely, in the nociceptive tail flick test, we scored a significant reduction of reactivity in *Eps15L1*-KO versus WT controls (Fig 1C). More importantly, we observed lack of feeding in 90% of *Eps15L1*-KO pups, as evidenced by the absence of milk in their stomach, which represents the probable cause of death of these animals (Fig 1C). In addition, we observed difficulties in respiratory activity at the second day after birth in most of the *Eps15L1*-KO pups (Fig 1C), which might be either a primary defect or secondary to malnutrition. Of note, *Eps15L1*-KO mice that escaped perinatal death showed reduced growth rate (Fig S2E and F), neurological deficits (Fig S2G), and succumbed at 6–8 wk. In contrast, we have previously published that EPS15-KO mice do not show any neurological abnormality (Pozzi et al, 2012).

In summary, EPS15L1—in contrast to EPS15—is required for life in mice, possibly in conjunction with an essential nonredundant function of EPS15L1 in the nervous system.

### Evidence for a nonredundant role of EPS15L1 in synaptic function

The sum of the above data and the lessons learned in model organisms (Salcini et al, 2001; Koh et al, 2007) prompted us to investigate a possible role for EPS15L1 in SVR. In the *Drosophila* model system, the ablation of *eps15* was accompanied by a sizable decrease in the levels of dynamin and intersectin (Majumdar et al, 2006; Koh et al, 2007). Thus, we initially assessed the levels of a panel of endocytic proteins in the brain of newborn *Eps15*-KO and *Eps15L1*-KO mice. We observed no differences in the levels of expression of most of the proteins, including the core components of the endocytic machinery: AP2, dynamin, and clathrin (Fig 2A). An ~50% decrease in the levels of intersectin-1 was detected in *Eps15L1*-KO mice (Fig 2A and B). This reduction appears to be specific for intersectin-1, as other presynaptic (e.g., synapsin, synaptophysin, and VGAT) or postsynaptic (e.g., GluR1) markers did not significantly change between *Eps15L1*-KO and *Eps15L1*-WT mice (Fig 2A and B). We concluded that the KO of either *Eps15L1* or *Eps15* does not have a general impact on the expression levels of the synaptic proteins.

Next, we performed an FM dye–based SVR assay on mature neurons cultured for 14 d in vitro. Fluorescence was measured after a first depolarization in the presence of the dye induced by KCl (F1,

Fig 2C, left) and then again after a second depolarization (F2, Fig 2C, left) to measure dye release from synaptic vesicles. We did not detect any significant difference in either F1 or F2, arguing for no apparent defects in SVR (Fig 2C, right). However, compensatory endocytic mechanisms might mask a defect in classical SVR (Nguyen et al, 2014); thus, we proceeded with an ultrastructural analysis of *Eps15L1*-KO neurons. We detected a reduction of synaptic vesicles in *Eps15L1*-KO synapses of about 50%, although synapses from *Eps15*-KO were comparable with WT (Fig 2D). The number of docked/tethered vesicles was also significantly decreased in *Eps15L1*-KO synapses (Fig 2E). This result is reminiscent of similar defects evidenced in model organisms (Salcini et al, 2001; Koh et al, 2007) and suggested that defective synaptic function may be responsible for the neurological deficits of *Eps15L1*-KO mice and possibly for neonatal lethality.

To test whether the absence of evident phenotypes in the dye uptake assay was due to the up-regulation of compensatory bulk endocytosis, we measured by EM the number of vesicles with a diameter higher than 80 nm, as bulk endocytosis is typically characterized by large invaginations of the plasma membrane which then fission to form endosomal-like compartments (Cousin, 2009; Saheki & De Camilli, 2012). Under steady state conditions, we did not observe differences in the number of this type of vesicles (Fig 2F). However, when we followed HRP uptake upon depolarization with 50 mM KCl (Fig 2G, left), we observed a significant increase in large HRP-positive structures in *Eps15L1*-KO neurons (Fig 2G, center and right), suggesting that bulk endocytosis is indeed more active in these cells.

### EPS15 and EPS15L1 are redundantly essential in embryonic development

To investigate possible organismal redundant roles for EPS15 and EPS15L1, we generated *Eps15*/*Eps15L1*-double KO (DKO) mice by inter-crossing *Eps15*-KO/*Eps15L1*-HET. DKO embryos died shortly after 9.5 d post coitum (dpc) (Fig 3A), and already at 9.5 dpc, they showed severe morphological defects. Some of them presented severe developmental delay; all embryos, instead, displayed reduced midbrain–hindbrain boundary, fused somites, absence of limb bud, and delayed turning of the heart. Moreover, DKO embryos appeared paler than their controls (Fig 3B), possibly because of reduced vascularization or a defect in hematopoiesis, or both. PECAM staining of the vascular system effectively evidenced a reduced vascularization both of the embryo proper as well as of the yolk sac (Fig 3C and D). A more detailed confocal analysis of the PECAM staining confirmed the reduced and compromised vascularization of the head, somites, and yolk sac (Fig 3E) of DKO embryos.

To further characterize the vascular phenotype, we generated mice in which *Eps15L1* was constitutively deleted and *Eps15* was conditionally deleted under the Tie2 promoter ($Eps15^{flp/flp}$/$Eps15L1^{-/-}$/$Tie2-Cre^{tg}$, referred to as conditional *Eps15*-KO/constitutive *Eps15L1*-KO, c15/L1KO). These mice displayed only a mild vascular defect (data not shown), suggesting that the severe phenotype observed in constitutive DKO mice was not due to a cell-autonomous defect of endothelial cells.

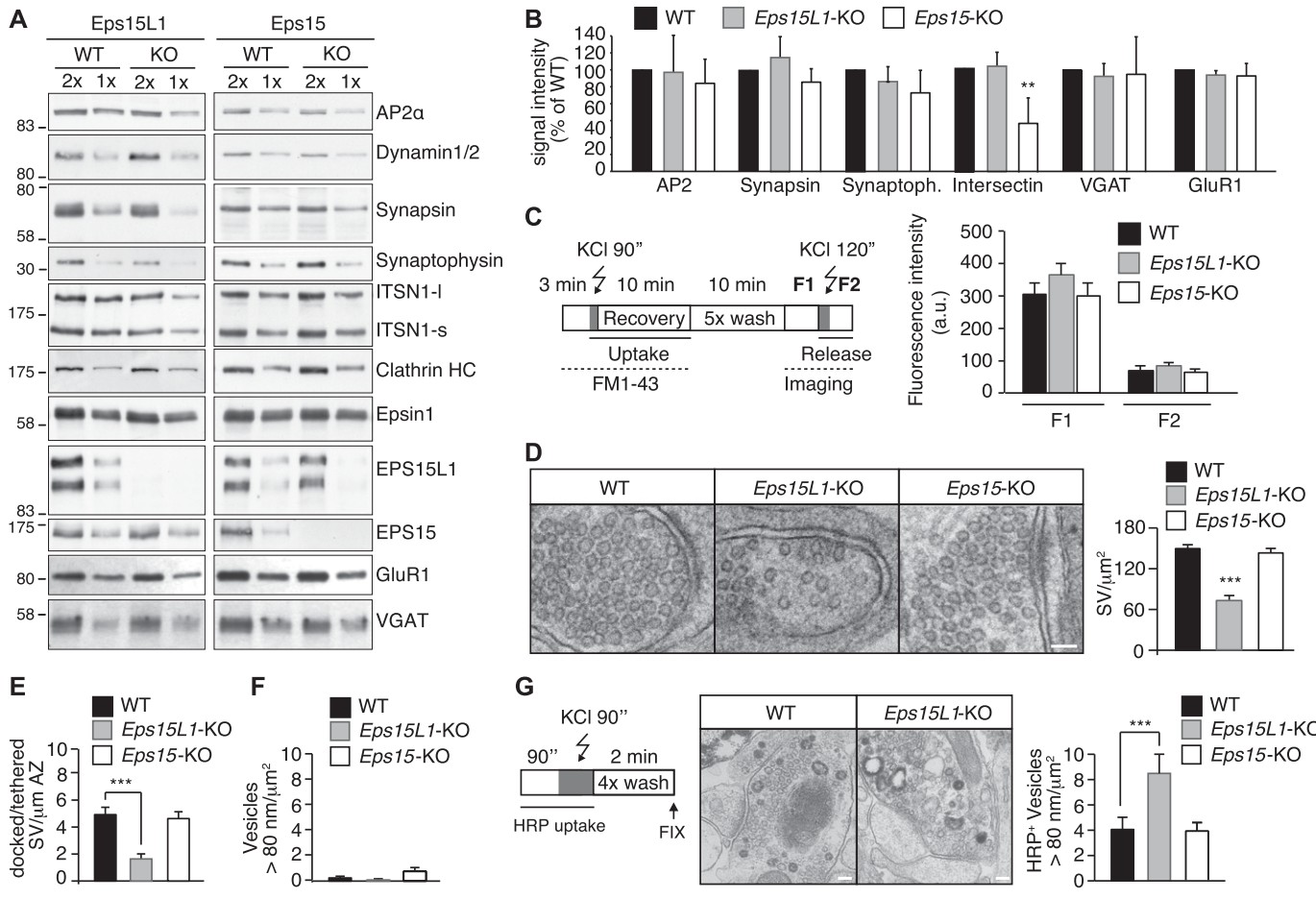

**Figure 2. EPS15L1 has a nonredundant role in neurons.**
**(A)** Western blotting of the indicated proteins in brain lysates from the indicated stains (neonatal mice). In the brain, two isoforms for intersectin-1 are present (ITSN1-l refers to the long isoform and ITSN1-s refers to the short isoform). Clathrin HC refers to clathrin heavy chain. Reduction of synaptophysin in *Eps15L1*-KO mice might be linked to a reduction in synapses, as observed in *Drosophila eps15* mutants (Koh et al, 2007); the reduction of intersectin-1, which directly interacts with EPS15L1, instead, might indicate a destabilization of the protein when it is not complexed. **(B)** A quantitation of the results shown in A is depicted, as obtained from replica experiments (n = 2–3). Results are the mean ± SD and are expressed as % of signal in WT sample, normalized for the loading control (vinculin). **(C)** Left, schematic representation of the experimental setup for FM dye uptake and release from hippocampal neurons. Grey boxes indicate KCl pulses. Right, averaged fluorescence intensity in WT, *Eps15L1*-KO, and *Eps15*-KO neurons after dye uptake (F1) and after depolarization with 50 mM KCl and release (F2). Differences among genotypes are not significant. **(D)** Left, ultrastructural analysis of steady-state synapses from WT, *Eps15L1*-KO, and *Eps15*-KO hippocampal neurons. Bar, 100 nm. Right, quantitation of synaptic vesicles (SV) per area by ultrastructural analysis. At least three different preparations of neurons were analyzed; ***P < 0.001 versus WT. **(E)** Quantitation of docked/tethered SV per length of active zone EM. At least three different preparations of neurons were analyzed; ***P < 0.001 versus WT. **(F)** Quantitation of vesicles with diameter higher than 80 nm (SV) per μm² by EM. At least three different preparations of neurons were analyzed. Differences among genotypes are not significant. **(G)** Left, schematic representation of the experimental setup for HRP uptake from hippocampal neurons. The grey box indicates the KCl pulse. Center, exemplary images by EM of HRP-labelled synapses from WT and *Eps15L1*-KO hippocampal neurons. Bar, 100 nm. Right, quantitation of HRP-positive (HRP⁺) vesicles with diameter higher than 80 nm (SV) per μm² by EM. ***P < 0.001 versus WT.
Source data are available for this figure.

## Deletion of *Eps15* and *Eps15L1* in the hematopoietic system affects maturation of RBCs

We sought to identify a cellular function redundantly regulated in a cell-autonomous fashion by EPS15 and EPS15L1 at the organismal level. We concentrated on erythrocyte maturation for a series of reasons: (i) the pallor of the *Eps15*/*Eps15L1*-DKO mice; (ii) the redundant impact of EPS15 and EPS15L1 on TfR internalization ([Huang et al, 2004] and Fig 4A); (iii) the known role of TfR internalization in erythrocyte maturation (Levy et al, 1999; Zhu et al, 2008; Ishikawa et al, 2015; Muckenthaler et al, 2017); and (iv) the availability of c15/L1KO mice.

First, we investigated the expression of EPS15 and EPS15L1 during erythrocyte maturation. Mice were treated with phenylhydrazine (PHZ) to induce hemolytic anemia and formation of immature RBCs (Fig S3A and B). Consistent with a role in maturation of RBCs, we observed high levels of expression of EPS15 and EPS15L1 at the peak of immature RBCs (7–10 d after PHZ treatment), which dramatically decreased at 16 d (Fig S3C). The same kinetics were observed for TfR, intersectin-1, and AP2, compatible with a role for all these proteins in determining the wave of TfR required for maturation of RBCs (Fig S3C).

We then turned to the c15/L1KO mice. Analysis of the peripheral blood of newborn c15/L1KO mice revealed a significant reduction

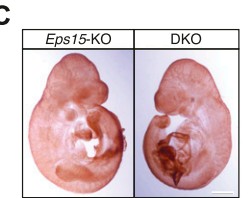

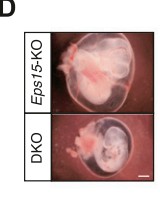

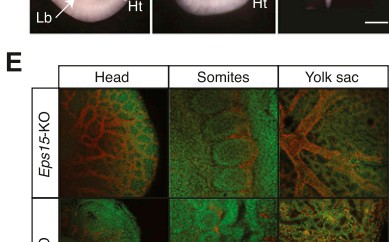

**A**

| dpc | litters | *Eps15*-KO/<br>*Eps15L1*-WT | *Eps15*-KO/<br>*Eps15L1*-HET | *Eps15*-KO/<br>*Eps15L1*-KO | total |
|---|---|---|---|---|---|
| 9.5 | 19 | 39 (27%) | 75 (53%) | 28 (20%) | 142 |
| 10.5 | 2 | 1 (7%) | 11 (78%) | 2 (14%) | 14 |
| 11.5–12.5 | 3 | 12 (41%) | 15 (52%) | 0 (0%)* | 29 |

* 2 dead embryos

**Figure 3. EPS15 and EPS15L1 are redundantly essential in embryonic development.**
**(A)** Absolute and percent numbers of live embryos of the indicated genotypes, observed between 9.5 and 12.5 dpc. **(B)** *Eps15*-KO and DKO fresh embryos at 9.5 dpc. Arrows point to the limb bud (Lb), the midbrain–hindbrain boundary (Bd) and the heart (Ht). The star and the continuous lines indicate, respectively, the absence of limb bud and the presence of fused somites in DKO embryos, as compared with *Eps15*-KO mice where somites are not fused (indicated with dotted line). Some DKO embryos had a more severe phenotype and reduced size than others. Bar, 1 mm. **(C)** *Eps15*-KO and DKO embryos at 9.5 dpc stained with anti-PECAM. Bar, 1 mm. **(D)** *Eps15*-KO and DKO embryos and yolk sacs at 9.5 dpc stained with anti-PECAM. Bar, 1 mm. **(E)** Confocal images of head, somites, and yolk sac from *Eps15*-KO and DKO embryos at 9.5 dpc, stained with anti-PECAM (red). Nuclei were counterstained with TOPRO (green). Bar, 100 $\mu$m.

of the mean corpuscular volume (MCV) and an increase in RBC distribution width, a value typically high in iron-deficient anemia (Fig 4B). Mice in which only *Eps15* was deleted under the *Tie2* promoter (*Eps15*$^{flp/flp}$/*Tie2-Cre*$^{tg}$, referred to as conditional *Eps15*-KO, c15KO) displayed a minor, if any, reduction in MCV, whereas constitutive single *Eps15L1*-KO (L1KO) had no phenotype. Finally, May–Grünwald–Giemsa staining revealed the presence of anisotropic RBCs and a significant increase in reticulocyte number specifically in c15/L1KO mice (Fig 4C and D).

To analyze adult hematopoiesis, we generated mice in which both *Eps15* and *Eps15L1* were conditionally deleted under the *Tie2* promoter (*Eps15*$^{flp/flp}$/*Eps15L1*$^{flp/flp}$/*Tie2-Cre*$^{tg}$, conditional *Eps15*/*Eps15L1*-DKO, referred to as cDKO). Western blotting analysis on major tissues (the brain, liver, and spleen) confirmed that *Eps15* and *Eps15L1* were deleted in *Tie2*[+] cells (Tang et al, 2010) of cDKO mice (Fig S3D). Adult cDKO mice were anemic as revealed by significant reduction of all analyzed parameters (Fig 4E). May–Grünwald–Giemsa staining revealed a great variation in the size and shape of cDKO RBCs (Fig 4F, top). Finally, o-dianisidine staining clearly showed that RBCs from cDKO were hypochromic (Fig 4F, bottom; additional characterizations of iron metabolism, in cDKO mice, are shown in Fig S3E and F). Thus, cDKO mice suffer from microcytic hypochromic anemia (Levy et al, 1999; Zhu et al, 2008; Ishikawa et al, 2015).

### EPS15 and EPS15L1 redundantly regulate TfR internalization in RBCs

In search of a molecular explanation for the RBC phenotype in cDKO mice, we concentrated on TfR endocytosis. We reasoned that the phenotype of cDKO (impaired maturation of RBCs and increased serum iron) could be explained by defective iron uptake through TfR endocytosis in RBCs of these mice. To discriminate mature RBCs from reticulocytes, we used thiazole orange (TO) to detect the nucleic acids in immature RBCs. Consistent with defective erythrocyte maturation, we detected twice as many reticulocytes (TO-positive cells, TO[+]) in the blood of cDKO (5.5 ± 0.4 in WT versus 10.7 ± 1.5 in cDKO, *P* < 0.001 [Fig 4G, top]). In addition, surface TfR

expression was retained in ~50% of mature RBCs (TO-negative cells, TO[–]) in cDKO mice, whereas it was virtually absent—as expected—in WT mice (Fig 4G, bottom, and H). Finally, in cDKO mice, a significantly higher fraction of TO[+] retained TfR surface expression versus the WT counterparts (Fig 4G, bottom, and H).

### Redundancy and uniqueness of EPS15 and EPS15L1

We have herein characterized a number of redundant and non-redundant functions of EPS15 and EPS15L1 at the organismal level in mice.

In our previous work, we showed that *Eps15*-KO mice are viable and fertile and, notwithstanding extensive phenotype screening, showed only a partial defect in B-cell lymphopoiesis (Pozzi et al, 2012). At variance, we show here that *Eps15L1*-KO mice show readily detectable neurological defects, associated with reduced number of synaptic vesicles, and neonatal lethality. We have not proved that the neonatal phenotype is a direct consequence of the neurological defect and of altered SVR. Neuronal-specific conditional KO will be needed to address this issue. However, we note that the depletion in synaptic vesicles detected in *Eps15L1*-KO mice is highly reminiscent of the phenotypes observed in the nematode and in flies, upon deletion of the sole orthologue of *Eps15* in these species. In these latter cases, it was established that the SVR phenotype was responsible for the neurological defects and, in the case of *Drosophila*, for lethality (Salcini et al, 2001; Koh et al, 2007).

Our studies also allowed the identification of a number of redundant functions of *Eps15* and *Eps15L1*. These two genes are required (redundantly) for correct embryo development. The severity and complexity of the phenotype argues for the possibility that not a single but a multitude of signaling pathways is affected in these mice: a possibility compatible with the role of these proteins in a pervading process such as endocytosis. At the phenotypic level, one major alteration, which might contribute importantly to the lethal phenotype of DKO embryos, was subverted angiogenesis. This phenotype, however, was non–cell autonomous, and its exact mechanism remains to be elucidated.

One redundant function that we were able to precisely dissect, both cellularly and molecularly, concerns erythropoiesis. Here, we

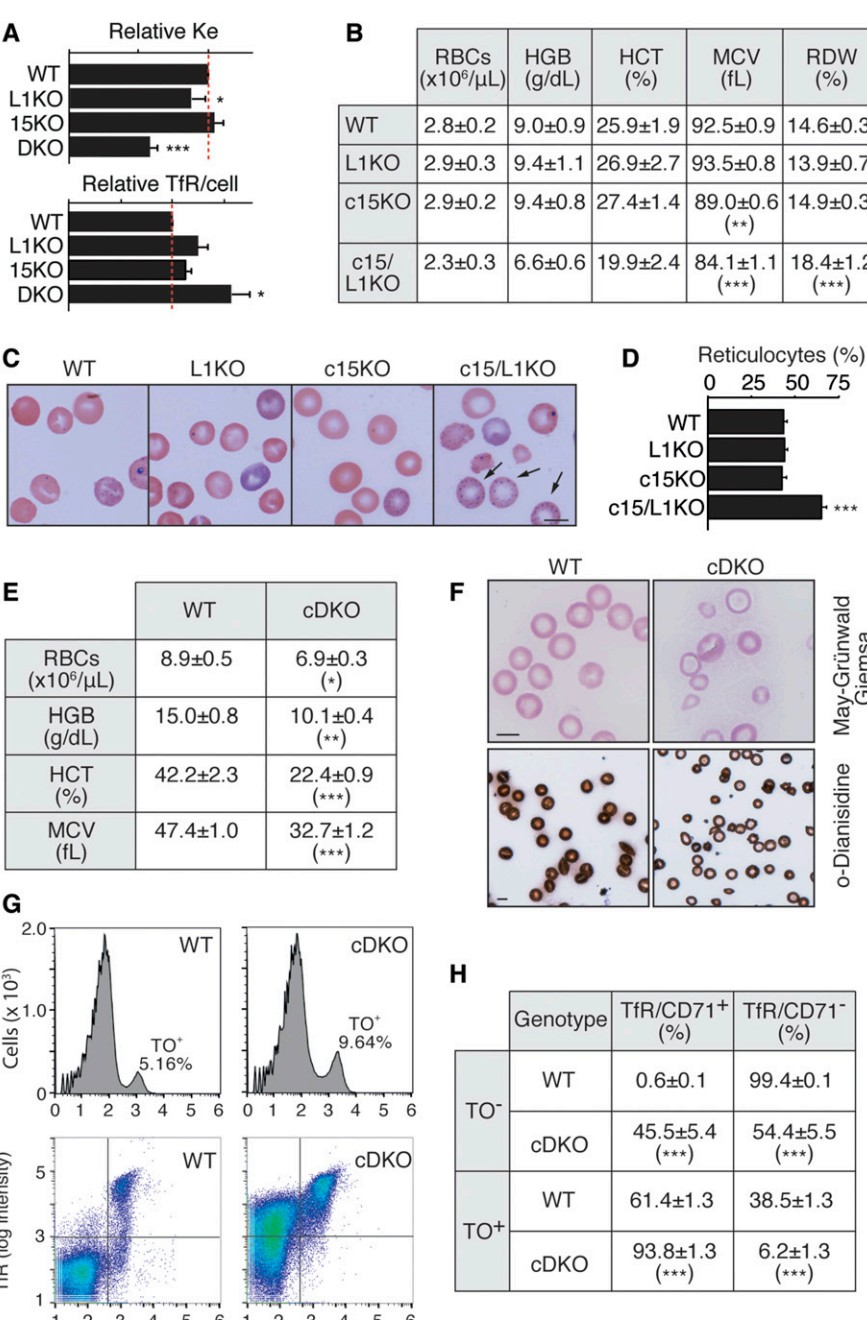

**Figure 4. Deletion of *Eps15* and *Eps15L1* in the hematopoietic system causes microcytic hypochromic anemia.**

**(A)** Relative TfR endocytic constant ($K_e$) (top) and TfR surface levels per cell (bottom) in WT, *Eps15L1*-KO (L1KO), *Eps15*-KO (15KO), and DKO fibroblasts. WT value (= 1.0) is shown by a dashed red line. **(B)** Values of RBCs, hemoglobin (HGB), hematocrit (HCT), MCV, and RBC distribution width in newborn WT, *Eps15L1*-KO (L1KO), conditional *Eps15*-KO (c15KO), and conditional *Eps15*-KO/Eps15L1-KO (c15/L1KO) mice. At least eight animals for each genotype were analyzed. **(C)** May–Grünwald–Giemsa staining of blood smears from newborn mice of the indicated genotype. Arrows point to reticulocytes, which retain methylene blue dye. Bar, 10 μm. **(D)** Quantitation of the experiments shown in **(C)**. Number of newborn mice analyzed: WT = 7, L1KO = 6, c15KO = 6, c15/L1KI = 6. **(E)** Values of RBCs, HGB, HCT, and MCV in adult WT and cDKO mice. At least nine animals of each genotype were analyzed. **(F)** Blood smears of adult WT and cDKO mice stained with May–Grünwald–Giemsa (top) or o-dianisidine (bottom). Bar, 10 μm. **(G)** Representative fluorescence-activated cell sorting (FACS) analysis of peripheral blood from adult WT and cDKO mice, stained with thiazole orange (TO) to reveal reticulocytes (top) or with TfR (CD71) and thiazole orange for nucleic acids (bottom). In the top panels, the percentage of TO+ cells in one representative WT and in one representative cDKO is shown. In the main test, average ± SEM of TO+ cells, calculated from at least five animals of each genotype, is indicated. **(H)** Percentage of positive and negative RBCs for surface TfR (CD71) among TO+ and TO− blood cells, in adult WT and cDKO mice. At least five animals of each genotype were analyzed. In (A), (B), (D), (E), and (H): *P < 0.05 versus WT; **P < 0.01 versus WT; ***P < 0.001 versus WT.

showed that EPS15 and EPS15L1 are redundantly required for the internalization of the TfR, which at the organismal level leads to microcytic hypochromic anemia when lacking in the hematopoietic system. These results are reminiscent of those obtained in hematopoietic-specific KO mice for PICALM, an accessory endocytic adaptor and binding partner of EPS15 and EPS15L1. These mice suffered from microcytic hypochromic anemia, displaying reduced RBCs, haemoglobin, haematocrit, and MCV, and increased reticulocyte number and serum iron (Ishikawa et al, 2015). Moreover, also in these mice, anemia was caused by defective TfR internalization in RBCs.

Evolutionarily, the duplication of the ancestral *Eps15* locus must have occurred in a vertebrate ancestor, within metazoan, because all vertebrates display two orthologues, whereas invertebrates possess only one (Fig S3G). By analyzing comparatively the results in invertebrates (Salcini et al, 2001; Koh et al, 2007) and in mammals ([Pozzi et al, 2012] and this study), it is likely that the essential functions of proto-EPS15 were retained by EPS15L1, whereas EPS15 diverged to assume more fine-tuning roles, while retaining a large (albeit not complete) spectrum of redundancy with EPS15L1. This possibility is supported by the

closer similarity of vertebrate EPS15L1s with the invertebrate orthologues and by the slower rate of divergence of this gene versus EPS15, during vertebrate radiation (Fig S3G). The high degree of similarity and collinearity between the two genes/proteins should now enable structure–function experiments to unmask the structural determinant responsible for the unique function of EPS15L1 in neurons.

## Materials and Methods

### Generation and maintenance of mouse strains

The *Eps15*-KO was previously described (Pozzi et al, 2012). The conditional *Eps15*-KO allele (*Eps15*$^{tm1a(KOMP)Wtsi}$, in the text referred to as *Eps15*$^{flp/flp}$) was obtained from the Sanger Institute. The PGK-neo cassette was removed by breeding to general deleter *Flp* mice and the *Flp* transgene was removed in subsequent breedings. The *Eps15L1*-KO was generated in a C57BL6 background by Ozgene, through insertion of a PGK-neo cassette next to the first coding exon. The first exon and the PGK-neo cassette were removed by crossing with general deleter *Cre* mice and the *Cre* transgene was removed in subsequent breedings. The conditional *Eps15L1*-KO allele (*Eps15L1*$^{flp/flp}$) was obtained from Polygene. The PGK-neo cassette was removed by breeding to general deleter *Flp* mice and the *Flp* transgene was removed by subsequent breedings.

For isolation of MEFs, *Eps15*$^{flp/flp}$ mice were crossed with *Eps15L1*$^{+/-}$ and *Rosa26*$^{CreERT2}$ mice (Ventura et al, 2007).

For hematological studies, *Eps15*$^{flp/flp}$ mice were crossed with *Eps15L1*$^{+/-}$ or *Eps15L1*$^{flp/flp}$ and *Tie2-Cre*$^{tg}$ mice (Kisanuki et al, 2001) to obtain deletion of the floxed genes in endothelial and hematopoietic cells.

Mice were kept on 12 h light/dark cycle with ad libitum access to water and food. Experiments were performed in accordance with our institution guidelines and the Italian Laws (Project numbers: 06/12 and 222/16).

### Behavioral test

Neonatal behavior tests were scored giving either a normal or abnormal evaluation to the observed phenotype. To evaluate neonatal motor and nociception functions, pups were subjected to the tail suspension and the tail flick test, respectively (Sternberg et al, 2004; Takahashi et al, 2010).

In the tail suspension test, the pup was gently held by the tip of the tail and the position observed recorded. A normal reaction was scored when the pup was hanging symmetrically with all four limbs wide open. The test was scored as abnormal if the hind limbs were touching or clasping during the test.

In the tail flick test, the pup was held between the thumb and forefinger in an upright position and the distal tip of the tail was gently lowered into a water bath maintained at 50°C. The latency to vigorous tail withdrawal was recorded, with a 15-s cutoff time. The test was repeated after a 30-s interval and the latencies were averaged. Tail withdrawl within 15 s was scored as normal, whereas longer retraction times were scored as abnormal.

The feeding status was estimated at the day of birth by carefully observing the pups for the presence of milk in the stomach, which is visible through the thin abdominal skin of pups.

The respiration behavior was assessed by visual examination. Apneas, sighs, gasps, paroxysmal, or periodic breathing were considered as pathological. All tests were performed in blind respect to the genotype of the animals.

### Primary culture of hippocampal neurons

Primary neurons were isolated from hippocampi dissected from P0 newborn mice and placed in cold HBSS buffer. Digestion of the hippocampi was performed at 37°C in water bath for 10 min using papain (20 U/ml final) in EBSS buffer with the addition of DNase (500 U/ml final) (all from Worthington Biochemical Corporation). After trituration of the solution, the cell pellet was washed two times in EBSS and resuspended in Neurobasal-A medium supplemented with B27, GlutaMAX, and antibiotics (all from Life Technologies). The cells were plated onto glass coverslips or glass bottom dishes (MatTek Corporation) coated with 1 mg/ml poly-D-lysine hydrobromide (Sigma-Aldrich). Primary cultures from hippocampal neurons were used at 14 d from their isolation.

### Dye uptake experiments

For dye uptake experiments, FM1-43 dye (Molecular probes from Life Technologies) was used at a concentration of 1 $\mu$m. Recovery buffer contained 130 mM NaCl, 5 mM KCl, 2 mM CaCl$_2$, 2 mM MgCl$_2$, 25 mM Hepes (pH 7.33), and 30 mM glucose. Stimulation buffer contained as above except for 85 mM NaCl and 50 mM KCl.

DL-2-amino-5-phosphonopentanoic acid (AP-V; Sigma-Aldrich), CNQX (Ascent Scientific), and sodium kynurenate (Ascent Scientific) were added to all solutions, except HBSS, at the final concentration of 50 $\mu$M, 100 $\mu$M, and 1 mM, respectively.

Dye uptake experiments of primary hippocampal neurons were performed at room temperature as follows: the cells were incubated 3 min in recovery buffer with dye and then stimulated with stimulation buffer with dye for 90 s, following 10 min in recovery buffer with dye. After 10 min of several washes in HBSS buffer (Life Technologies), neurons were bathed for 2 min in HBSS with 10 mM ADVASEPT (Sigma-Aldrich) and washed two times with HBSS alone. For acquisition of microscope images, neurons were placed in recovery buffer (without dye) and stimulated with stimulation buffer (without dye).

Fluorescence microscopy was performed on an UltraVIEW VoX (Perkin Elmer) spinning disk confocal unit, equipped with an Eclipse Ti inverted microscope (Nikon), a C9100-50 emCCD camera (Hamamatsu), and driven by Volocity software (Improvision, Perkin Elmer). Images were acquired with a 60× oil immersion objective (NA 1.4) as Z-stacks (0.3 $\mu$m step). F1 stacks (dye uptake) were acquired while neurons were in the recovery buffer. F2 stacks (after dye release) were taken after 2 min from the application of the stimulation buffer.

Images were analyzed using ImageJ software (ImageJ 1.43u) as follows: F1 and F2 Z-stacks were transformed in single images using the max intensity tool and subsequently concatenated.

The resulting F1-F2 stack was aligned and the background was subtracted. Fluorescence intensity (F1 and F2) of single punctuated signals was automatically calculated using a homemade macro.

## Electron microscopy and morphometry

Conventional electron microscopy was carried out as previously described (Polishchuk et al, 1999; Beznoussenko et al, 2007). Hippocampal neuron cultures were directly fixed with 1% glutaraldehyde in 0.2 M Hepes (pH 7.2–7.4). The number of vesicles (i.e., synaptic vesicles, docked/tethered or vesicles larger than 80 nm) was determined on electron micrographs with ImageJ software (ImageJ 1.43u) and a homemade ImageJ script. Synaptic vesicles were defined as the small homogenously sized vesicles (≤60 nm) forming large clusters in the terminal. Docked/tethered vesicles were defined as small homogenously sized vesicles (≤60 nm) within 25 nm from the active zone PM. For bulk steady-state endocytosis, we measured vesicles larger than 80 nm in the terminal. For HRP uptake experiments, peroxidase from horseradish Type VI-A (Sigma-Aldrich) was used at concentration of 10 mg/ml. All buffers with inhibitors were the same as for the "dye uptake experiments" described above. Neurons were allowed to equilibrate for 90 s in the recovery buffer with HRP, and then stimulated for 90 s with the stimulation buffer with HRP. After four quick washes with the recovery buffer, the neurons were fixed with 2.5% glutaraldehyde/0.1 M cacodylate, pH 7.4.

Images were analyzed using ImageJ software (ImageJ 1.43u). The number of HRP-positive vesicles larger than 80 nm was automatically calculated using a homemade ImageJ Script. Statistical significance was calculated by $t$ test.

## Whole-mount staining of *Eps15/Eps15L1*-DKO embryos with anti-PECAM primary antibody

Embryos were collected at 9.5 dpc and fixed overnight at 4°C in 100% methanol. After fixation, the embryos were rehydrated and blocked overnight at 4°C in PBS containing 5% donkey serum, 1% BSA, and 0.5% Triton-X 100. After wash, they were incubated overnight at 4°C with anti-PECAM primary antibody (kindly provided by E. Dejana) in PBS containing 0.5% BSA and 0.25% Triton-X 100. The primary antibody was revealed by VECTASTAIN ABC system (Vectorlabs), according to the manufacturer's instructions, and images acquired under a Leica stereo microscope. Alternatively, PECAM was revealed by immunofluorescence as follows: embryos were incubated with anti-rat Alexa 488 and TOPRO (Thermo Fisher Scientific) in PBS containing 0.5% BSA and 0.25% Triton-X 100 for 2 h at room temperature. The embryos were then washed, post-fixed in 4% paraformaldehyde, and mounted with ProLong Gold. Images were taken with a Leica TCS SP5 confocal microscope.

## Isolation of MEFs

MEFs were isolated from $Eps15^{flp/flp}/Eps15L1^{+/+}/CreERT2^{tg}$ and $Eps15^{flp/flp}/Eps15L1^{-/-}/CreERT2^{tg}$ embryo mice. Briefly, pregnant mice were euthanized at 13.5 dpc by $CO_2$ asphyxia. The embryos were decapitated for genotyping and the internal organs removed.

The embryos were then dissociated with a blade, transferred to 0.05% trypsin plus 0.02% EDTA solution, and incubated for 10 min at 37°C under agitation. After addition of 100 µg/ml DNase I (Roche), the embryos were incubated for further 10 min. Digestion was completed by passing embryonic tissues through an 18-gauge needle. The cells were finally isolated through a 70-µm cell strainer and plated in 10-cm petri dishes. The cells were grown in Hepes buffered GlutaMAX-DMEM (Gibco Invitrogen) supplemented with 10% fetal bovine serum (HyClone), at 37°C and 9% $CO_2$.

## Radioactive internalization and saturation binding assay

Radioactive internalization and saturation binding assay were performed as previously described (Sigismund et al, 2005). Experiments were performed on WT, *Eps15L1*-KO, *Eps15*-KO, and *Eps15/Eps15L1*-DKO MEFs. WT and *Eps15*-KO fibroblasts were derived from $Eps15^{flp/flp}/Eps15L1^{+/+}/CreERT2^{tg}$ mice, after in vitro treatment with DMSO vehicle or 250 nM (Z)-4-hydroxytamoxifen (Sigma-Aldrich), respectively. *Eps15L1*-KO and *Eps15/Eps15L1*-DKO fibroblasts were derived from $Eps15^{flp/flp}/Eps15L1^{-/-}/CreERT2^{tg}$ mice, after in vitro treatment with DMSO vehicle or 250 nM (Z)-4-hydroxytamoxifen, respectively. At least three different preparations of cells were analyzed.

## Biochemical studies

Dissection of the brains for immunofluorescence and fractionation of the brains for Western blotting was performed as previously described (Offenhauser et al, 2006).

Western blotting on whole tissue extracts and RBC lysates was performed as follows. Tissues (isolated after sacrifice of the mice by $CO_2$ asphyxia) and peripheral RBCs (collected as explained in the next section) were washed in PBS and homogenized in a lysis buffer containing 50 mM Hepes, pH 7.4, 150 mM NaCl, 1% glycerol, 1% triton X-100, 1.5 mM $MgCl_2$, 5 mM EGTA, 1 mM PMSF, 10 mM sodium orthovanadate, 50 mM sodium fluoride, and protease inhibitor cocktail SetIII (Calbiochem). The lysates were then clarified at 4°C by centrifugation at 120,000 $g$ for 1 h (for tissues) or 10,000 $g$ for 15 min (for RBCs). Protein concentration was measured by BCA protein assay (Thermo Fisher Scientific), according to the manufacturer's instructions. Desired amounts of proteins were dissolved in Laemmli buffer (final concentration: 2% SDS, 50 mM Tris–HCl, pH 6.8, 100 mM DTT, 10% glycerol, and 0.001% bromophenol blue), boiled for 5 min, separated by SDS–PAGE, and then transferred on nitrocellulose membranes. The membranes were blocked for 1 h in 5% BSA or 5% low-fat dry milk (in TBS plus 0.1% Triton X-100) and then incubated overnight at 4°C with the primary antibody. After wash, the membranes were incubated with HRP-conjugated secondary antibody (Cell Signaling) for 1 h at room temperature. The bound secondary antibody was revealed through chemiluminescence by photographic films (Amersham Hyperfilm ECL) or under a ChemiDoc Imaging System, after incubation with ECL substrate. Quantitation of the blots was performed using ImageJ (ImageJ 1.43u). The following primary antibodies were used: anti-α adaptin (mouse monoclonal AP6; Thermo Fisher Scientific), anti-clathrin heavy chain (rabbit polyclonal #4796; Cell Signaling),

anti-dynamin1/2 (mouse monoclonal #MABT188; EDM Millipore), homemade anti-EPS15 (mouse monoclonal 3T3), homemade anti-EPS15L1 (rabbit polyclonal #860), anti-epsin-1 (mouse monoclonal ZZ3, kindly provided by S. Polo), anti-intersectin-1 (rabbit polyclonal #499, kindly provided by S. Polo), anti-synapsin (rabbit polyclonal #A6442; Thermo Fisher Scientific), anti-PSD95 (mouse monoclonal #MA1-046; Affinity Bioreagents), anti-synaptophysin (mouse monoclonal #101011 and rabbit polyclonal #101002 Synaptic Systems), anti-GluR1 (rabbit polyclonal, #06-306 Upstate), anti-VGAT (rabbit polyclonal, #131013 Synaptic Systems), anti-TfR (mouse monoclonal #H68.4; Thermo Fisher Scientific), anti-tubulin (mouse monoclonal #T5168; Sigma-Aldrich), and anti-vinculin (mouse monoclonal #V9131; Sigma-Aldrich).

### Collection and staining of blood smears

To analyze protein expression during induced erythropoiesis, WT mice were treated with 1% PHZ (Sigma-Aldrich) in sterile PBS at day 0 (4 ml/g) and at day 3 (6 ml/g). The animals were then monitored daily, and blood was collected at different time points, as reported in the text. For all hematological studies, the blood was collected after decapitation or by tail vein puncture. During collection, blood was transferred in an EDTA-containing solution at 100 mM final concentration. The blood was analyzed in a hemocytometer (Beckman Coulter) and smeared on glass slides. For determination of reticulocytes, the blood smears were stained with reticulocyte stain (Sigma-Aldrich), based on methylene blue dye, or with May–Grünwald–Giemsa stain (Sigma-Aldrich), according to the manufacturer's instructions. For determination of intracellular hemoglobin, the blood smears were stained with o-dianisidine (Sigma-Aldrich), as described in Fibach & Prus (2005). After staining, the blood smears were air-dried and examined under a Leica stereo microscope, using a 100× oil immersion objective.

### FACS staining of RBCs

After collection and dilution in an EDTA-containing solution, peripheral RBCs were washed with cold PBS and blocked for 1 h in PBS plus 10% BSA at 4°C. The cells were then incubated overnight at 4°C with anti-CD71 antibody conjugated to PE (BD) or APC (eBioscience) and diluted in PBS plus 1% BSA. After wash, the cells were incubated with thiazole orange (BD) for 1 h at room temperature. The samples were acquired on a FACSCanto II (Becton Dickinson). Data were analyzed with FlowJo 10.1 software.

### Analysis of iron metabolism

Serum iron levels were determined by MULTIGENT Iron assay (Abbott), according to the manufacturer's instructions. Serum transferrin and ferritin levels were determined with Transferrin Mouse ELISA kit (Abcam) and Ferritin Mouse ELISA kit (Abcam), respectively, according to the manufacturer's instructions.

### Perls' Prussian blue staining

After sacrifice of the mice by $CO_2$ asphyxia, the liver and spleen were isolated, washed in PBS, and fixed overnight in 4% paraformaldehyde. After paraffin-embedding, 3-$\mu$m sections were cut. Staining was performed using Perls' Prussian blue stain kit (DDK Italia), according to the manufacturer's instructions. Examination was performed by digital imaging, using the Leica Aperio ScanScope scanning system. Representative images were extrapolated at 20× optical zoom.

### Statistical analysis

For each experiment, the number of observations (number of animals or cellular culture preparations) is specified within figures or reported in the figure legends. All values are average ± SEM, and differences were analyzed to detect statistical significance with a two-tailed $t$ test (n.s., not significant; *$P < 0.05$; **$P < 0.01$; ***$P < 0.001$).

### Study approval

All animal studies were conducted with the approval of Italian Minister of Health (03/2008; 06/2012; 222/16) and were performed in accordance with the Italian law (D.lgs. 26/2014), which enforces Dir. 2010/63/EU (Directive 2010/63/EU of the European Parliament and of the Council of 22 September 2010 on the protection of animals used for scientific purposes).

## Supplementary Information

## Acknowledgements

We thank Rosalind Gunby for critically reading the manuscript. We thank Dr Simona Polo (IFOM, Milan) for generously providing antisera against intersectin. We are grateful to Alberto Gobbi, Emanuela Capillo, and the Mouse facility (Cogentech Società Benefit Srl, Milan). We thank the Imaging facility at IFOM/Cogentech Società Benefit Srl, Milan, and the ALEMBIC facility at San Raffaele Scientific Institute, Milan. We thank the Centre European of Nano-medicine (CEN, Italy) for the possibility to use Tecnai 20 electron microscope. This work was supported by grants from the Worldwide Cancer Research (16-1245) to S Sigismund; Associazione Italiana per la Ricerca sul Cancro (AIRC IG 18988 and MCO 10.000) and the Italian Ministry of Health to PP Di Fiore; and the Italian Ministry of University and Scientific Research to C Tacchetti and PP Di Fiore (PRIN 2015XS92CC).

### Author Contributions

C Milesi: data curation and investigation.
P Alberici: investigation.
B Pozzi: investigation.
A Oldani: methodology.
GV Beznoussenko: methodology.
A Raimondi: methodology.
BE Soppo: methodology.
S Amodio: methodology.
G Caldieri: data curation and investigation.

MG Malabarba: data curation and project administration.
G Bertalot: methodology.
S Confalonieri: data curation and methodology.
D Parazzoli: methodology.
AA mironov: methodology.
C Tacchetti: supervision and methodology.
PP Di Fiore: conceptualization, supervision, and funding acquisition.
S Sigismund: conceptualization, data curation, supervision, project administration, and writing—original draft, review, and editing.
N Offenhäuser: conceptualization, data curation, supervision, project administration, and writing—original draft, review, and editing.

## Conflict of Interest Statement

The authors declare that they have no conflict of interest.

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
