## [Reviewer comments · Life Science Alliance]

Life Science Alliance

Redundant and non-redundant organismal functions of EPS15 and EPS15L1

Cinzia Milesi, Paola Alberici, Benedetta Pozzi, Amanda Oldani, Galina Beznoussenko, Andrea Raimondi, Blanche Ekalle Soppo, Stefania Amodio, Giusi Caldieri, Maria Grazia Malabarba, Giovanni Bertalot, Stefano Confalonieri, Dario Parazzoli, alexandre mironov, Carlo Tacchetti, Pier Paolo Di Fiore, Sara Sigismund, and Nina Offenhäuser

DOI: 10.26508/lsa.201800273

Corresponding author(s): Sara Sigismund, IEO, Istituto Europeo di Oncologia IRCCS and Nina Offenhäuser, IFOM, Fondazione Istituto FIRC di Oncologia Molecolare

Review Timeline:

Submission Date:	2018-12-10
Editorial Decision:	2018-12-10
Revision Received:	2019-01-14
Editorial Decision:	2019-01-14
Revision Received:	2019-01-15
Accepted:	2019-01-16

Scientific Editor: Andrea Leibfried

Transaction Report:

Please note that the manuscript was previously reviewed at another journal and the reports were taken into account in inviting a revision for publication at *Life Science Alliance* prior to submission to *Life Science Alliance*.

Reviewer #1 Review

Comments to the Authors (Required):

This short Ms reports on the generation and phenotypic characterization of novel constitutive and conditional single and double knockout (KO) mice lacking the endocytic adaptor proteins Eps15 and Eps15L1. It is shown that single loss of Eps15L1 in mice causes perinatal lethality with incomplete penetrance that correlates with neurological defects, reduced presynaptic protein levels and reduced synaptic vesicle numbers suggestive of a crucial function of Eps15L1 in presynaptic function. In contrast, Eps15/ Eps15L1 DKO leads to embryonic lethality around E9.5 that is accompanied by developmental and vascular defects as well as partial anemia. The latter phenotype likely is a consequence of impaired endocytosis of the transferrin receptor and corresponding iron uptake defects arising from a redundant role of Eps15 and Eps15L1 in developing red blood cells akin to loss of PICALM, another early-acting endocytic adaptor protein.

The description of redundant and unique roles of the only two Eps15 family members in mice clearly is of interest to the community. The data contained in the paper are solid and of high quality, although they remain at a somewhat superficial level of characterization of the KO phenotype as explained below. I recommend publication of the paper in this journal once some of the mechanistic aspects of this study have been strengthened.

#1-4 The observation of a possible neuron-specific function of Eps15L1 seems intriguing but at this stage remains poorly understood. Several open issues arise that in my view would need to be addressed prior to publication:

1. Expression levels of pre- and postsynaptic proteins: A single non-quantified immunoblot is shown that suggests reduced synaptophysin and intersectin levels in Eps15L1 KO brains. In order to better understand the phenotype quantitative data from several mice and a careful analysis of both pre- and postsynaptic proteins of excitatory and inhibitory synapses is required to really interpret the results.
2. The suggested reduced expression of synaptophysin may either reflect specific presynaptic defects or reduced synapse numbers. The latter should be analyzed at the level of brain slices and cultured neurons using available pre- and postsynaptic markers.
3. At least some effort should be made to find out whether synaptic transmission is impaired. This could be done either electrophysiologically or using pHluorin assays in cultured neurons.
4. EM: Does loss of Eps15L1 affect synapse or active zone size or the number of docked SVs? Are there signs of abnormally sized or shaped vesicles? This is not overt from the single images provided.

#5. Vascular defects: Is there any change in the level or localization of the VEGFR, another known CME substrate?

#6. The overlapping role of Eps15 and Eps15L1 in developing RBCs strikingly resembles the phenotype of PICALM loss. Are there changes in the level or distribution of CALM in cells from DKO mice?

#7. Is the observed defect in TfR endocytosis in absence of Eps15 /Eps15L1 a reflection of impaired endocytic pit formation at the PM of reticulocytes or a specific defect in cargo recognition of the TfR, which, however, at least to my knowledge has not been shown to

associate with either Eps15 or Eps15L1?

Minor point: It is stated in the text that only 10% of the expected Eps15L1 pups are alive at weaning. However, looking at the figure it seems that 10% of all weaned pups are KO (instead of the expected 25%).

Reviewer #2 Review

Comments to the Authors (Required):

The manuscript by Milesi et al analyzes the respective roles of eps15L1 and eps15 in vivo in the context of the mouse. Unlike eps15 knockouts, eps15L1 knockouts are 90% perinatally lethal and display neurological defects, consistent with synaptic enrichment of Eps15L1. The synaptic defects were relatively mild, with normal FM-dye uptake and a 50% reduction in synaptic vesicles, as judged by EM. This section of the analysis would be enhanced by additional synapse functional assays, such as electrophysiology. Further analysis in eps15/eps15L1 double mutants showed synthetic embryonic lethality with reduced vasculature, among other defects. Following this, the authors focus on a redundant role for eps15 and eps15L1 in hematopoiesis and transferrin metabolism. Overall the experiments described represent a lot of work, appear very well done, and the conclusions are clear and in general not overstated. The main issue here is the degree to which this work advances the field. The finding that eps15L1 displays mild synaptic defects, while roles in development and transferrin uptake are redundant with eps15, does not seem to advance the field very far beyond what was already known from invertebrate models.

Reviewer #3 Review

Comments to the Authors (Required):

Milesi et al. have studied the in vivo roles of EPS15 and EPS15L1 in knockout mice models. Previous work has shown that Eps15 KO mice are viable and fertile, and cell biological experiments have indicated that EPS15 and EPS15L1 may have redundant functions in clathrin-mediated endocytosis. Here, the authors have generated EPS15L1 KO mice and found that these showed neonatal lethality, albeit with incomplete penetrance. These mice showed signs of impaired synaptic functions which could be attributed to a 50% reduction in synaptic vesicle numbers as detected by electron microscopy of synapses. EPS15-EPS15L1 double KO mice were generated, and these showed early embryonic lethality, demonstrating the redundant roles of EPS15 and EPS15L1. Conditional KO of EPS15 and EPS15L1 during adult hematopoiesis resulted in a phenotype consistent with microcytic hypochromic anemia. This could be mechanistically explained by reduced Transferrin receptor endocytosis in the red blood cells of these mice.

Overall this is a thorough and high-quality study that sheds light on the in vivo functions of two proteins of great importance in endocytosis. As such, the study is of potential interest for the readership of this journal. I have no technical concerns with this manuscript, but I am a little uncertain whether the study provides a sufficient conceptual advance for this journal since we already knew that EPS15 and EPS15L1 are important for transferrin receptor endocytosis with possibly redundant functions, and since EPS15 has already been studied in vivo before, in mice, nematodes and flies.

December 10, 2018

Re: Life Science Alliance manuscript #LSA-2018-00273-T

Dr. Sara Sigismund
IFOM
Via Adamello 16
Milan 20139
Italy

Dear Dr. Sigismund,

Thank you for transferring your manuscript entitled "Redundant and non-redundant organismal functions of EPS15 and EPS15L1" to Life Science Alliance. The manuscript was assessed by expert reviewers at another journal before, and the editors transferred those reports to us with your permission.

The reviewers who evaluated your work elsewhere appreciated that EPS15L1 affects synaptic vesicle numbers/density non-redundantly with EPS15, and that both proteins function redundantly to regulate transferrin receptor endocytosis, thereby modulating erythrocyte maturation. While they found the data overall convincing, they expected much further reaching insight. This is not needed for publication in Life Science Alliance. I would thus like to invite you to provide a revised version of your work for publication here. Only a minor revision is needed. We would expect a point-by-point response to the criticisms raised as well as adding quantitative data for pre- and postsynaptic proteins (Reviewer #1, pt 1) and a re-analysis of the data already at hand to address reviewer #1's point 4 (analysis of EM data).

The typical timeframe for revisions is three months.

Thank you for this interesting contribution to Life Science Alliance. We are looking forward to receiving your revised manuscript.

Sincerely,

Andrea Leibfried, PhD
Executive Editor
Life Science Alliance
Meyerhofstr. 1

69117 Heidelberg, Germany
t +49 6221 8891 502
e a.leibfried@life-science-alliance.org
www.life-science-alliance.org

- A letter addressing the reviewers' comments point by point.
- An editable version of the final text (.DOC or .DOCX) is needed for copyediting (no PDFs).
- High-resolution figure, supplementary figure and video files uploaded as individual files: See our detailed guidelines for preparing your production-ready images, <http://life-science-alliance.org/authorguide>
- Summary blurb (enter in submission system): A short text summarizing in a single sentence the study (max. 200 characters including spaces). This text is used in conjunction with the titles of papers, hence should be informative and complementary to the title and running title. It should describe the context and significance of the findings for a general readership; it should be written in the present tense and refer to the work in the third person. Author names should not be mentioned.

B. MANUSCRIPT ORGANIZATION AND FORMATTING:

Full guidelines are available on our Instructions for Authors page, <http://life-science-alliance.org/authorguide>

Reviewer 1

This short Ms reports on the generation and phenotypic characterization of novel constitutive and conditional single and double knockout (KO) mice lacking the endocytic adaptor proteins Eps15 and Eps15L1. It is shown that single loss of Eps15L1 in mice causes perinatal lethality with incomplete penetrance that correlates with neurological defects, reduced presynaptic protein levels and reduced synaptic vesicle numbers suggestive of a crucial function of Eps15L1 in presynaptic function. In contrast, Eps15/Eps15L1 DKO leads to embryonic lethality around E9.5 that is accompanied by developmental and vascular defects as well as partial anemia. The latter phenotype likely is a consequence of impaired endocytosis of the transferrin receptor and corresponding iron uptake defects arising from a redundant role of Eps15 and Eps15L1 in developing red blood cells akin to loss of PICALM, another early-acting endocytic adaptor protein.

The description of redundant and unique roles of the only two Eps15 family members in mice clearly is of interest to the community. The data contained in the paper are solid and of high quality, although they remain at a somewhat superficial level of characterization of the KO phenotype as explained below. I recommend publication of the paper in this journal once some of the mechanistic aspects of this study have been strengthened.

R. We thank the reviewer for his/her positive comment on our work. We have now addressed the main issues raised by this reviewer, as detailed below.

#1-4 The observation of a possible neuron-specific function of Eps15L1 seems intriguing but at this stage remains poorly understood. Several open issues arise that in my view would need to be addressed prior to publication:

- 1. Expression levels of pre- and postsynaptic proteins: A single non-quantified immunoblot is shown that suggests reduced synaptophysin and intersectin levels in Eps15L1 KO brains. In order to better understand the phenotype quantitative data from several*
- 2. The suggested reduced expression of synaptophysin may either reflect specific presynaptic defects or reduced synapse numbers. The latter should be analyzed at the level of brain slices and cultured neurons using available pre- and postsynaptic markers. mice and a careful analysis of both pre- and postsynaptic proteins of excitatory and inhibitory synapses is required to really interpret the results.*

R. In the revised version of the manuscript, we have included in Fig. 2B a quantitation of pre-synaptic and post-synaptic markers from 2-3 mice. In Fig. 2A, B, we have also included a post-synaptic marker (GluR1, excitatory synapse) and a pre-synaptic marker (VGAT, inhibitory synapse). From these analyses, we could observe a significant decrease in the level of Intersectin1 in *Eps15L1*-KO mice as compared to WT mice (see new Fig. 2A, B). This reduction appears to be specific for Intersectin1, as the other markers did not significantly change among *Eps15L1*-KO and -WT mice (new Fig. 2A, B). We concluded that the KO of either *Eps15L1* or *Eps15* does not have a general impact on the expression levels of the synaptic proteins (main text lines 26-32).

3. *At least some effort should be made to find out whether synaptic transmission is impaired. This could be done either electrophysiologically or using pHluorin assays in cultured neurons.*

R. This is an important suggestion and we are definitely interested in pursuing these studies in the future. At this stage, however, we believe that they would entail an effort that exceeds the scope and the aim of the present study.

4. *EM: Does loss of Eps15L1 affect synapse or active zone size or the number of docked SVs? Are there signs of abnormally sized or shaped vesicles? This is not overt from the single images provided.*

R. We could not score any significant difference in the active zone size among WT and *Eps15L1*-KO mice. Conversely, when we measured by EM the number of docked/tethered vesicles, we found that they are significantly decreased in *Eps15L1*-KO mice vs. WT (new Fig. 2E). By HRP uptake assay after depolarization with 50 mM KCl, we also observed a significant increase in large HRP-positive structures (larger than 80 nm) in *Eps15L1*-KO neurons (new Fig. 2G). We concluded (Pag. 7, lines 13-20) that bulk endocytosis is indeed more active in *Eps15L1*-KO neurons, possibly to compensate the defect in the number of synaptic vesicles.

#5. *Vascular defects: Is there any change in the level or localization of the VEGFR, another known CME substrate?*

R. We have performed a preliminary analysis of VEGFR levels and activation by western blot in MEFs derived from KO mice. However, since this initial analysis revealed no substantial differences, we have not pursued this direction further.

#6. *The overlapping role of Eps15 and Eps15L1 in developing RBCs strikingly resembles the phenotype of PICALM loss. Are there changes in the level or distribution of CALM in cells from DKO mice?*

R. We did not check CALM levels.

#7. *Is the observed defect in TfR endocytosis in absence of Eps15 /Eps15L1 a reflection of impaired endocytic pit formation at the PM of reticulocytes or a specific defect in cargo recognition of the TfR, which, however, at least to my knowledge has not been shown to associate with either Eps15 or Eps15L1?*

R. On the basis of extant literature, the more probable hypothesis is that *Eps15* and *Eps15L1* recognize the TfR indirectly, via the clathrin adaptor AP2. Indeed, they possess different AP2-binding motifs and they interact with AP2 with high stoichiometry (Iannolo G. et al., 1997; Coda L., 1998). In addition, they have been shown to play a critical role in initial steps of CCP formation, recruiting AP2 and cargo (Taylor MJ et al., 2011; Henne WM et al., 2010; Ma L. et al., 2016). Whether they exert a specific and direct role in TfR recognition has not been reported, and our data do not allow to reach this conclusion.

Minor point: It is stated in the text that only 10% of the expected Eps15L1 pups are alive at weaning. However, looking at the figure it seems that 10% of all weaned pups are KO (instead of the expected 25%).

R. The reviewer is right and we apology for the mistake: 10% of all weaned pups are KO (instead of the expected 25%). We have corrected the text accordingly.

Reviewer 2

The manuscript by Milesi et al analyzes the respective roles of eps15L1 and eps15 in vivo in the context of the mouse. Unlike eps15 knockouts, eps15L1 knockouts are 90% perinatally lethal and display neurological defects, consistent with synaptic enrichment of Eps15L1. The synaptic defects were relatively mild, with normal FM-dye uptake and a 50% reduction in synaptic vesicles, as judged by EM. This section of the analysis would be enhanced by additional synapse functional assays, such as electrophysiology. Further analysis in eps15/eps15L1 double mutants showed synthetic embryonic lethality with reduced vasculature, among other defects. Following this, the authors focus on a redundant role for eps15 and eps15L1 in hematopoiesis and transferrin metabolism. Overall the experiments described represent a lot of work, appear very well done, and the conclusions are clear and in general not overstated. The main issue here is the degree to which this work advances the field. The finding that eps15L1 displays mild synaptic defects, while roles in development and transferrin uptake are redundant with eps15, does not seem to advance the field very far beyond what was already known from invertebrate models.

R. See our reply below, after reviewer 3 comments.

Reviewer 3

Milesi et al. have studied the in vivo roles of EPS15 and EPS15L1 in knockout mice models. Previous work has shown that Eps15 KO mice are viable and fertile, and cell biological experiments have indicated that EPS15 and EPS15L1 may have redundant functions in clathrin-mediated endocytosis. Here, the authors have generated EPS15L1 KO mice and found that these showed neonatal lethality, albeit with incomplete penetrance. These mice showed signs of impaired synaptic functions which could be attributed to a 50% reduction in synaptic vesicle numbers as detected by electron microscopy of synapses. EPS15-EPS15L1 double KO mice were generated, and these showed early embryonic lethality, demonstrating the redundant roles of EPS15 and EPS15L1. Conditional KO of EPS15 and EPS15L1 during adult hematopoiesis resulted in a phenotype consistent with microcytic hypochromic anemia. This could be mechanistically explained by reduced Transferrin receptor endocytosis in the red blood cells of these mice.

Overall this is a thorough and high-quality study that sheds light on the in vivo functions of two proteins of great importance in endocytosis. As such, the study is of potential interest for the readership of this journal. I have no technical concerns with this manuscript, but I am a little uncertain whether the study provides a sufficient conceptual advance for this

journal since we already knew that EPS15 and EPS15L1 are important for transferrin receptor endocytosis with possibly redundant functions, and since EPS15 has already been studied in vivo before, in mice, nematodes and flies.

R. The evaluations of the study made by Reviewer 2 and 3 were similar. Both reviewers agreed that the work is carefully performed and technically of high quality. However, they both made the issue of conceptual advance, since a neuronal function for *Eps15/L1* has been previously described in nematodes and flies. We respectfully submit that:

i) The role of *Eps15* and *Eps15L1* was never characterized in mammals *in vivo*. The fact that the mammalian proteins play a role in the nervous systems, as in the case of the invertebrate ones, was not obvious before our study;

ii) Our data show that mammalian *Eps15L1* orthologue has specifically retained this neuronal function, at variance with *Eps15*.

iii) More importantly, the redundant role of *Eps15* and *Eps15L1* in erythropoiesis and iron metabolism was never described before.

Our data help to clarify how *Eps15L1* has retained the ancestral role in the nervous system, while the two mammalian proteins have acquired a new redundant function in erythropoiesis.

January 14, 2019

RE: Life Science Alliance Manuscript #LSA-2018-00273-TR

Dr. Sara Sigismund
IFOM
Via Adamello 16
Milan 20139
Italy

Dear Dr. Sigismund,

Thank you for submitting your revised manuscript entitled "Redundant and non-redundant organismal functions of EPS15 and EPS15L1". I appreciate the introduced changes and I am thus happy to publish your paper in Life Science Alliance.

Before sending you the official acceptance letter, please log into our system one more time to fill in the electronic license to publish form. Your manuscript number will change to LSA-2018-00273-TRR, please make sure to populate this new manuscript number with all manuscript files (single click process).

Please log in to your account: <https://lsa.msubmit.net/cgi-bin/main.plex>

A. FINAL FILES:

-- High-resolution figure, supplementary figure and video files uploaded as individual files: See our detailed guidelines for preparing your production-ready images, <http://life-science-alliance.org/authorguide>

B. MANUSCRIPT ORGANIZATION AND FORMATTING:

Full guidelines are available on our Instructions for Authors page, <http://life-science-alliance.org/authorguide>

Sincerely,

January 16, 2019

RE: Life Science Alliance Manuscript #LSA-2018-00273-TRR

Dr. Sara Sigismund
IEO, Istituto Europeo di Oncologia IRCCS
Via Adamello 16
Milan 20139
Italy

Dear Dr. Sigismund,

Thank you for submitting your Research Article entitled "Redundant and non-redundant organismal functions of EPS15 and EPS15L1". It is a pleasure to let you know that your manuscript is now accepted for publication in Life Science Alliance. Congratulations on this interesting work.

DISTRIBUTION OF MATERIALS:

Again, congratulations on a very nice paper. I hope you found the review process to be constructive and are pleased with how the manuscript was handled editorially. We look forward to future exciting submissions from your lab.

Sincerely,
